# A Kinetic Study of the Pozzolanic Reaction of Fly Ash, CaO, and Na_2_O in the Preparation of Fly Ash Belite Cement

**DOI:** 10.3390/ma12203303

**Published:** 2019-10-11

**Authors:** Yongfan Gong, Ben Yu, Yonghao Fang, Dingyi Yang, Shu-an Wu, Yurong Yan

**Affiliations:** 1Departments of Civil Science and Engineering, University of Yangzhou, Yangzhou 225127, China; yuben19970326@163.com (B.Y.); ydy1991@163.com (D.Y.); 2College of Mechanics and Materials, University of Hohai, Nanjing 210098, China; 3Departments of Building Engineering, University of Yangzhou Polytechnic, Yangzhou 225009, China; wsan2019@163.com (S.-a.W.); zzl5868@yeah.net (Y.Y.)

**Keywords:** low activity, fly ash, belite cement, hydrothermal reaction, NaOH, kinetic model

## Abstract

Fly ash belite cement is a kind of low-carbon cement prepared by a two-step process involving hydrothermal synthesis and low-temperature calcination. Pozzolanic reaction pastes, as the precursors of fly ash belite cement prepared by hydrothermal synthesis, are affected mainly by reaction temperature, time, ratios of the mass of fly ash/lime (FA/CA), and the dosage of Na_2_O. The absorbance rate of CaO with reaction time was tested for all samples, and the reaction kinetic model and parameters of the granule-hydrothermal synthesis method were discussed. A kinetic model for the hydrothermal synthesis in the presence of Na_2_O was proposed based on the Kondo’s modified Jander equation and Arrhenius equation. The activation energy (Ea) of the process was determined to be 67.76 kJ/mol. In addition, with an increasing dosage of Na_2_O, the pre-exponential factor A of the Arrhenius equation increased. However, the hydrothermal reaction degree was accurately predicted using the kinetic model characterized by the absorption rate of CaO. The results indicated that Na_2_O, as an alkali activator, facilitated the diffusion of Ca^2+^ firstly, then partly dissolved the amorphous phase in the mixtures and, finally, accelerated the formation of poorly crystallized hydrates.

## 1. Introduction

Nowadays, the annual output of fly ash is approximately 600 million tons in China. A majority is used effectively, and fly ash-based materials have been widely studied. However, some types, obtained by the wet process, have been not employed in basic building materials because of their low pozzolanic activity [1,2,3,4]. An effective way of using these fly ash resources is in the development of fly ash belite cement (FABC). The preparation process of FABC is divided into two steps [5,6,7,8,9,10], namely hydrothermal synthesis and low-temperature calcination. In the hydrothermal synthesis process, the raw materials of fly ash and lime are mixed uniformly, followed by hydrothermal synthesis at a set temperature ranging from 97 °C to 250 °C. In the low-temperature calcination process, the precursors as pozzolanic reaction drying products are calcined at a set temperature ranging from 700 °C to 950 °C. Finally, active dicalcium silicon (2CaO·SiO_2_; β-C_2_S) and calcium aluminate (12CaO·7Al_2_O_3_; C_12_A_7_) are formed, and some unhydraulic inertia materials, such as mullite, quartz, and iron oxide, still exist. Recently, some researchers have proposed that some activators such as NaOH, KOH, CaSO_4_, and Na_2_SO_4_ can accelerate the hydrothermal synthesis process and improve the activity of FABC. Guerrero’s group introduced a slurry-autoclaving synthesis process, whereby the slurries of fly ash and lime with high water/solid ratios (5:1–10:1) were prepared [11]. Fang and Gong [12,13] suggested improving the activity of fly ashes in hydrothermal synthesis at 97 °C by the granule-hydrothermal synthesis method and chemical activation. Different from slurry, the mixed materials should be prepared as granules in order to reduce the heat consumption for drying synthesized precursors. According to previous studies [14,15], the hydrothermal synthesis process was considered as a kinetic model for pozzolanic reaction. The content of activated silicon and aluminum in fly ash was used to express the reaction rate using excess lime because the reaction of fly ash and lime always occurs in a stoichiometric ratio according to the chemical reaction equation [16], and the reaction rate can be expressed as the rate of change in any component. Shi [17,18] considered that the addition of CaCl_2_ and Na_2_SO_4_ had a significant effect on the reaction period, which could be applied to the Kondo’s modified Jander equation, as shown in Equation (1). When the conversion rate of Ca(OH)_2_ was expressed as the reaction degree α, the results showed that the reaction was accelerated by Na_2_SO_4_ and CaCl_2_ activation at curing temperatures of 23 °C, 35 °C, 50 °C, and 65 °C.
(1)$$F(\alpha )={\left[1-{\left(1-\alpha \right)}^{1/3}\right]}^{N}=kt$$
where α is the reaction degree, *N* is the reaction constant, *K* is the reaction constant, and *t* is the reaction time. 

Renedo and Fernandez [19,20] proposed a kinetic model of a pozzolanic reaction in the preparation of desulfurant sorbents, as shown in Equation (2). The reaction degree of Ca(OH)_2_ was expressed as x, and the weight ratios of fly ash versus Ca(OH)_2_ (FA/CA) were 3/1, 1/3, 3/5, and 11/12. All of them were activated by 0.5%, 1%, 1.5%, and 2% CaSO_4_·2H_2_O, respectively. The results showed that the reaction temperature T, the reaction time t, the mass ratio of FA/CA, the dosage of CaSO_4_·2H_2_O, and molar gas constant R were identified as Equation (2) parameters, and x as the reaction degree was influenced mostly by these experimental conditions. It proved that the shrinking core model fits the pozzolanic reaction of fly ash and lime well.
(2)$$t\times \left[(-378759{\mathrm{CaSO}}_{4}+3\times 10^{6})\times \frac{\mathrm{Flyash}}{\mathrm{Ca}{\left(\mathrm{OH}\right)}_{2}}+647386\right]\times \mathrm{EXP}\left(-\frac{57700}{\mathrm{RT}}\right)=1-{\left(1-x\right)}^{1/3}$$

However, it has been a long time since this research on the kinetic model of the pozzolanic reaction. Studies [21,22,23,24] have focused mainly on the reaction mechanism of fly ash during the hydration of cement. The kinetic study of pozzolanic reaction in the hydrothermal synthesis was lacking. Especially, the effect of an alkali activator on the pozzolanic reaction kinetics of hydrothermal synthesis was never reported. This study aimed to determine the kinetic expression of the hydrothermal synthesis reaction between fly ash and lime in the presence of Na_2_O based on the Kondo’s modified Jander equation and using the granule-hydrothermal synthesis method. The kinetic equation can explain the effect of temperature, time, FA/CA, and dosage of Na_2_O on the conversion of CaO. Meanwhile, it is possible to optimize the preparation process of precursors using the hydrothermal synthesis of fly ash belite cement [25,26].

## 2. Materials and Methods 

### 2.1. Raw Materials

Fly ashes were stored in the Xingtai Thermal Power Plant in Xingtai city of Northern China since the 1980s. The method of ultra-fine grinding was employed to increase the contact area of fly ash and lime—the fly ash and CaO (AR) were used as reactant materials, and NaOH (AR) was used as the Na_2_O activator to promote the hydrothermal synthesis process. The chemical composition of the high-carbon fly ash, which belongs to the third-grade FA by the Official Chinese National Standards (GB/T 1596-2017), is shown in Table 1.

### 2.2. Preparation of Specimens 

Previously, fly ashes and lime were mixed at a ratio of 4/1, 3/1, 7/3, and 13/7 by a mix machine. If the calcium source was CaO, it was put in a closed container for 2 h for lime digestion and to avoid carbonization. Na_2_O was added as equivalent NaOH and taken as the percentage of the total mass of CaO and fly ash. the dosage of Na_2_O was 0, 0.5%, 1.0%, 1.5%, respectively. The water (containing Na_2_O) to mixture ratio ranged from 0.3 to 0.4 so that the mixtures can be formed granules. The reactant materials were prepared as particles with a diameter of 8–12 mm diameter using a small-disk granulator under normal pressure. Then, the hydrothermal treatment was carried out at 57 °C, 77 °C, and 97 °C ranging from 0 to 40 h by a thermostat water bath (HH-4). 

### 2.3. Conversion of CaO

Thermal analyses were commonly used to determine the free CaO content in hydrothermal products by a muffle furnace(SX-10-13), which was recorded with the mass at 400 °C, 500 °C, 600 °C, 800°C, 950 °C using 500 mg samples at a heating rate of 10 °C/min and 30 min of soaking time. The mass of samples at different temperature heating was weighed with an electronic analytical balance. Ca(OH)_2_ and CaCO_3_ were considered as containing unreacted CaO (CaO_free_) calculated using Equation (3). The conversion of CaO (CaO_absorbed_) in the precursors was calculated using Equation (4) based on the following: decomposition of Ca (OH)_2_ between 400 °C and 500 °C (mass loss represented as Δm1), decomposition of CaCO_3_ between 600 °C and 800 °C (mass loss represented as Δm_2_), total mass loss at 950 °C (represented as LOI), loss on ignition of FA, and initial FA/CA. To remove the influence of carbon content, the computation of the mass loss was revised.
(3)$${\mathrm{CaO}}_{\mathrm{free}}(\%)=(\frac{\Delta m_{1}}{18}+\frac{\Delta m_{2}}{44})\times \frac{56}{100-\mathrm{LOI}}\times 100\%$$
(4)$${\mathrm{CaO}}_{\mathrm{absorbed}}(\%)=\left(1-\frac{{\mathrm{CaO}}_{\mathrm{free}}}{{\mathrm{CaO}}_{\mathrm{initial}}}\right)\times 100\%$$

## 3. Results

### 3.1. Analysis of Kinetic Characteristics

When the hydrothermal synthesis temperature was set as 97 °C, the CaO absorbed rates of precursors cured at 2, 4, 6, 8, 12, 16, 20, 24, 32, and 40 h are shown in Table 2. The results indicated that the FA/CA and dosage of Na_2_O had a significant impact on the CaO absorbed rate, which was consistent with the research results [11] and [12]. With the decrease in the FA/CA, the CaO absorbed rates decreased dramatically because of the higher contents of CaO containing more calcium hydroxide. With the addition of Na_2_O, the CaO absorbed rate of the precursors increased with the reaction time. This was principally because the OH^−^ ions were in favor of breaking the Si–O–Si (Al) bonds and effectively activating the pozzolanic property of fly ash. Substituting the CaO absorbed rate *α* and reaction time *t* into Equation (1), an apparent positive linear relationship was obtained between the value of ln[1 − (1 − α)^1/3^]^N^ and lnt. The slopes and intercepts were obtained by linear fitting based on the experimental data. The slopes were calculated using the reciprocal of the reaction grade N, and the intercepts were counted using (ln k)/N. The reaction grade N and reaction constant k were solved separately, and the fitting curve was obtained as shown in Table 3.

Based on the Jander equation model, the silica from fly ash was assumed as a constant-size spherical particle. The results of the reaction grade N ranging from 1 to 2 illustrate that the reaction process was controlled by the diffusion of the silica from fly ash converting into a porous layer. It was certain that the addition of Na_2_O significantly accelerated the granule-hydrothermal reaction behavior, however, the increase in the FA/CA decelerated it. When the content of CaO changed from 20% to 35% without Na_2_O addition, the reaction constant k reduced from 1.29 × 10^−2^ to 0.85 × 10^−2^. When the precursors containing 30% CaO were dosed with 0%, 0.5%, 1.0% and 1.5% Na_2_O, the reaction constant k was 0.88 × 10^−2^, 0.88 × 10^−2^, 0.96 × 10^−2^, and 1.19 × 10^−2^, respectively. This might be because the alkali activator accelerated the formation of poorly crystallized hydrates and partly dissolved the amorphous phase in the mixtures. Theoretically, the OH^−^ ions destroyed the crystal structure, spurred the activity of silica, and increased the dissolution of the amorphous phase in the reactants. The purpose was to increase the reaction reactivity, shorten the reaction time, and enhance the reaction rate greatly.

Figure 1 depicts the fitting curve obtained using the model versus the plot of the CaO absorbed rate obtained using the test at different times. From the start to 40 h, the variation trend of the fitting curves was very similar. As the dosage of Na_2_O was increased from 1.0 % to 1.5%, the CaO absorbed rates of reactants with different FA/CA became closer. Therefore, if the FA/CA was lower than 7/3, the Na_2_O-activation effect might be apparent compared with other conditions. To summarize, the yields predicted using the fitting curve were in good agreement with the experimental data of the CaO absorbed rate, which conformed to the actual development process accurately.

### 3.2. Kinetic Expression of a Hydrothermal Synthesis Reaction

The CaO absorbed rates of F3^a^ and F3^b^ in the hydration reaction at 57 °C and 77 °C are shown in Table 2. The reaction constant k at different hydrothermal synthesis temperatures was calculated by substituting the CaO absorbed rate α and reaction time t into Equation (1), as shown in Table 3. The results showed that the reaction constant k of F3 in the hydration reaction at 97 °C, 77 °C, and 57 °C was 0.881 × 10^−2^, 0.249 × 10^−2^, and 0.061 × 10^−2^, respectively. The reaction constant k for the mixed control model was also obtained using the Arrhenius equation expression, as shown in Equation (5). For the linear relationship between ln k and 1/T, the slope was −Ea/R. Therefore, the activation energy Ea in the process was determined to be 67.76 kJ/mol, and the correlation coefficient of this fitting curve (R^2^) was 0.9976, as shown in Figure 2.
(5)$$k=A\times \mathrm{EXP}\left(-\frac{\mathrm{Ea}}{\mathrm{RT}}\right)$$

Although the FA/CA changed, the reaction path and steps were not transformed. Therefore, the activation energy of this reaction was calculated previously using FA/CA to express the pre-exponential A [A = a(FA/CA) + b]. A fitting straight line was obtained by making a plot of the pre-exponential A versus FA/CA using the temperature values of 390 K, 370 K, and 350 K and the k values of 1.29 × 10^−2^, 1.07 × 10^−2^, 0.88 × 10^−2^, and 0.85 × 10^−2^, respectively, in this test. Parameters such as slope and intercept in the Arrhenius equation were calculated and are shown in Figure 3. A fair linear relationship was observed between the FA/CA and pre-exponential A.

The results showed that the slope was 7.98 × 10^6^ and the intercept was 1.53 × 10^7^, obtained by fitting straight lines. The reaction kinetic equation was established using the activation energy Ea (67.76 kJ/mol), and the reaction grade N (1.69) was achieved using the average N of F1, F2, F3, and F4, as shown in Equation (6).
(6)$$t\times \left[(7.98\times 10^{6})\times \frac{\mathrm{FA}}{\mathrm{CA}}+1.53\times 10^{7}\right]\times \mathrm{EXP}\left(-\frac{67760}{\mathrm{RT}}\right)={\left[1-{\left(1-\alpha \right)}^{1/3}\right]}^{1.69}$$
where FA/CA is the mass of fly ash to lime, T is the reaction temperature, t is the reaction time, R is the molar gas constant, and α is reaction degree.

By integrating Equations (1) and (5), the kinetic expression for the experimental process was obtained, which included the influence of FA/CA, reaction temperature, and time. This kinetic equation expression was used to predict the reaction constant k of precursors in the hydrothermal synthesis process. A comparison of the calculated data with the test data is shown in Table 4. The results indicated that the standard deviation obtained using the kinetic expression model was less than 3%. In addition, it was proven that this model had many advantages, including preferable prediction and wide application.

### 3.3. Kinetic Expression in the Presence of Na_2_O

In the Arrhenius equation, the pre-exponential factor A was influenced by collision frequency. In the presence of Na_2_O, OH^−^ ions could accelerate the dissolution of the amorphous phase in the hydrothermal synthesis of hydrates. With the increase in pre-exponential factor A, the collision frequency between Ca^2+^ ions and silicon-oxygen ions increased. The dosage of Na_2_O was set as the correction of the parameter FA/CA. The correction factor formulas are shown in Equation (7) and Equation (8).
(7)$$A=a\times \frac{\mathrm{FA}}{\mathrm{CA}}+1.53\times 10^{7}=k\times \mathrm{EXP}\left(\frac{\mathrm{Ea}}{\mathrm{RT}}\right)$$
(8)$$a=\frac{k\times \mathrm{EXP}(\mathrm{Ea}/\mathrm{RT})-1.53\times 10^{7}}{\mathrm{FA}/\mathrm{CA}}$$

To assess the value of a, the data, such as reaction constant k, FA/CA, and Ea, were substituted in the Equation (8)—the values are shown in Table 5. It was assumed that the value of a changed with the dosage of Na_2_O. The results showed that the value of a with 0%, 0.5%, 1%, and 1.5% Na_2_O was 7.98 × 10^7^, 8.11 × 10^6^, 9.21 × 10^6^, and 12.26 × 10^6^, respectively. This indicated that the value of a increased with the increase in Na_2_O doses. However, the fitting curve of the relationship between the value of a and the dosage of Na_2_O is shown in Figure 4.

According to the fitting curve, an exponential relationship as $a=a_{0}+fe^{-}{}^{\frac{Na_{2}O}{t}}$ existed between the value of a and the dosage of Na_2_O. It indicated that values of a_0_, f, and t were 7.78 × 10^6^, 1.31 × 10^5^, and −0.43423, respectively. The reaction kinetic Equation (9) was obtained by substituting these data into Equation (6).
(9)$$t\times \left[(7.78\times 10^{6}+1.31\times 10^{5}e^{\frac{Na_{2}O}{0.43423}})\times \frac{\mathrm{FA}}{\mathrm{CA}}+1.53\times 10^{7}\right]\times \mathrm{EXP}\left(-\frac{67760}{\mathrm{RT}}\right)={\left[1-{\left(1-\alpha \right)}^{1/3}\right]}^{1.69}$$
where FA/CA is the mass of fly ash to lime, T is the reaction temperature, *Na*_2_*O* is the dosage of Na_2_O, t is the reaction time, R is the molar gas constant, and *α* is reaction degree.

The reaction degree of CaO in the hydrothermal synthesis process in the preparation of FABC could be predicted at low reaction time using Equation (9). As NaOH was an active compound in the hydrothermal synthesis reaction, the optimal parameter designs could be achieved. The α values obtained from experiments and forecast using Equation (9) are shown in Figure 5. The predictive values were basically consistent with the experimental values. The average absolute value of the comparative error between the experimental and predictive values was less than 20%. Compared with the findings of Renedo [19,20], the results of the present study had an acceptable margin of error. The results were more accurate because the use of Na_2_O content as the control parameter of A benefited the improvement in the accuracy of predictive data. Meanwhile, the kinetic model was validated, and the findings had an important theoretical significance.

## 4. Discussion

The present study confirmed that the presence of Na_2_O accelerated the pozzolanic reaction between fly ash and lime in hydrothermal synthesis. OH^−^ ions were known to disrupt the vitreous body of fly ash and improve the formation rates of C–S–H gels. They are useful for the manufacture of FABC because they increase production efficiency and reduce the cost of production greatly. This significant stimulation of reaction kinetics in the presence of Na_2_O might be due to the combined effect of two factors.

### 4.1. Effect of Alkali Activation

The alkali Na^+^ accelerated the formation of poorly crystallized calcium silicate hydrates [27,28,29] and calcium aluminate hydrates, as a network modifier for the structure. The OH^−^ ions attacked and broke the Si–O–Si (Al) bonds, and then partly dissolved the amorphous phase to the mixtures. Hence, the Ca^2+^ ions were incorporated to generate more precursors and increase reactivity.

### 4.2. Effect of Higher Collision Frequency

When the zeta potential declined in the presence of Na^+^, it entailed a rise in ionic strength. Hence, the repulsion forces between ions were weakened by greater ionic strength [30,31], which was beneficial for the interparticle interaction by improving collision frequency. The pre-exponential factor A was affected by only the collision frequency of particles and increased with the the dosage of Na_2_O. This explained the steeper rise in hydration kinetics of hydrothermal synthesis in the presence of Na_2_O. This was the reason why the absorption rate of CaO increased with Na_2_O content.

## 5. Conclusions

This study investigated the reaction kinetics and optimization of the granule hydrothermal synthesis method with the addition of Na_2_O at atmospheric pressure. The conclusions and recommendations based on the experimental results were as follows:(1)Na_2_O, as a network modifier for the structure, incorporated the Ca^2+^ ions to generate more precursors and increase the reactivity. When the dosage of Na_2_O was ranging from 0% to 1.5%, the acceleration by alkali was more remarkable with the increase of dosage of Na_2_O.(2)The kinetic analyses based on the conversion of CaO indicated that the reaction process was controlled by the Kondo’s modified Jander equation. The trend of reaction in the presence of Na_2_O was very similar. The activation energy Ea in the process was determined to be 67.76 kJ/mol. It was inferred that the conversion of CaO values increased with the increase in the FA/CA.(3)A kinetic model for the hydrothermal reaction of lime–fly ash in the presence of Na_2_O was proposed, which could accurately predict the hydrothermal reaction degree as characterized by the absorption rate of CaO. NaOH could accelerate the dissolution of the amorphous phase in hydrothermal synthesis hydrates, and the dosage of Na_2_O was set as the correction of the parameter FA/CA. It was possible to increase the pre-exponential factor A in the presence of Na_2_O, which resulted in an increase in collision frequency between Ca^2+^ ions and silicon-oxygen ions.

## Figures and Tables

**Figure 1 materials-12-03303-f001:**
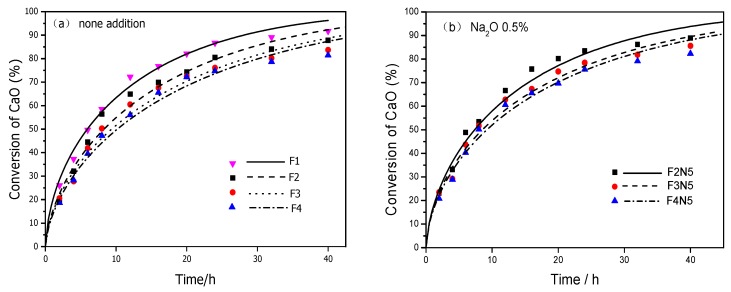
Experimental and simulated absorption rates of CaO under different conditions: (**a**) None addition; (**b**) the dosage of Na_2_O is 0.5%; (**c**) the dosage of Na_2_O is 1.0%; (**d**) the dosage of Na_2_O is 1.5%.

**Figure 2 materials-12-03303-f002:**
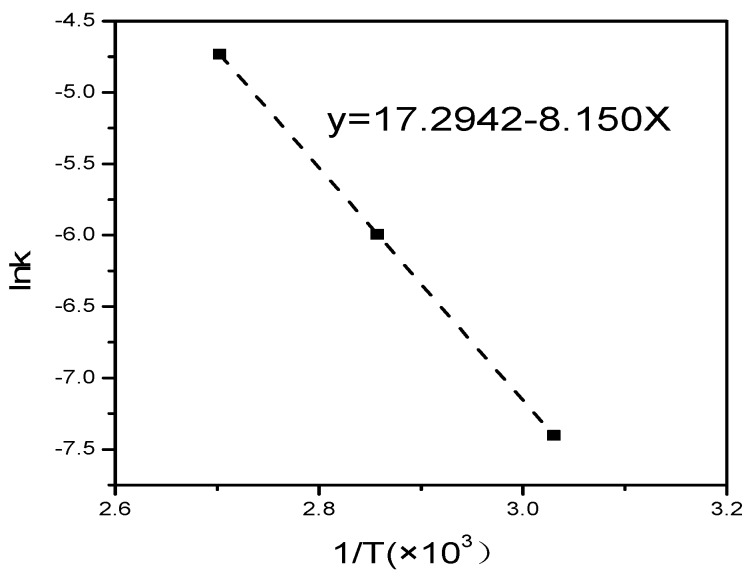
Linear relationship curve between ln k and 1/T.

**Figure 3 materials-12-03303-f003:**
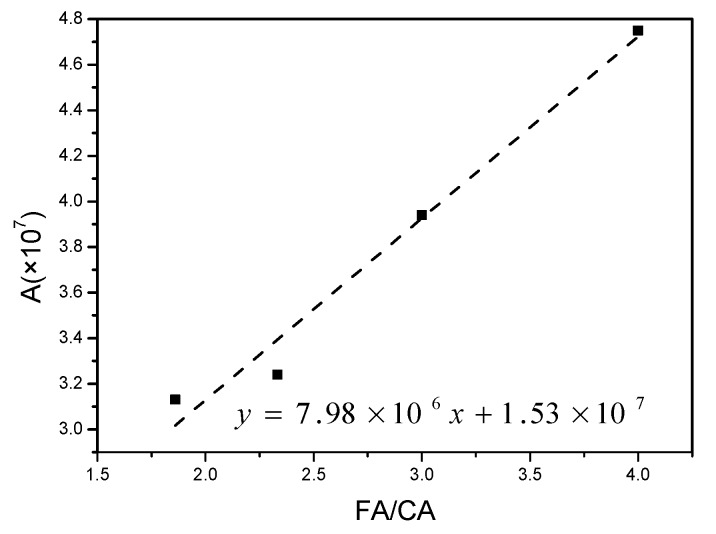
Linear relationship curve between A and FA/CA.

**Figure 4 materials-12-03303-f004:**
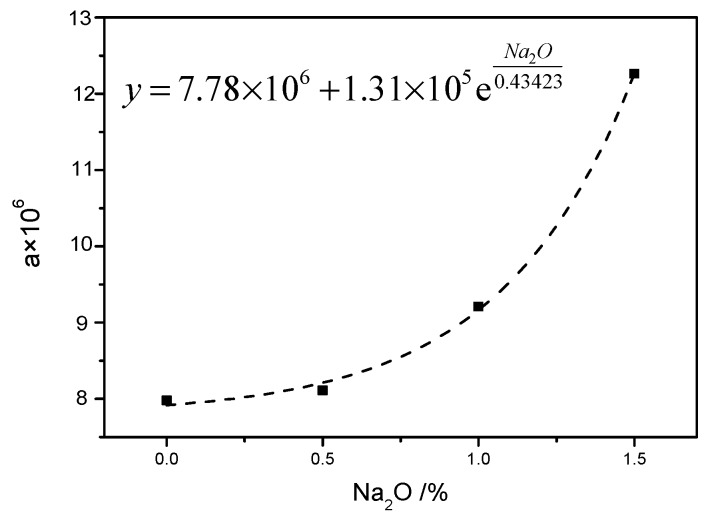
Relationship curve between values of a and dosages of Na_2_O.

**Figure 5 materials-12-03303-f005:**
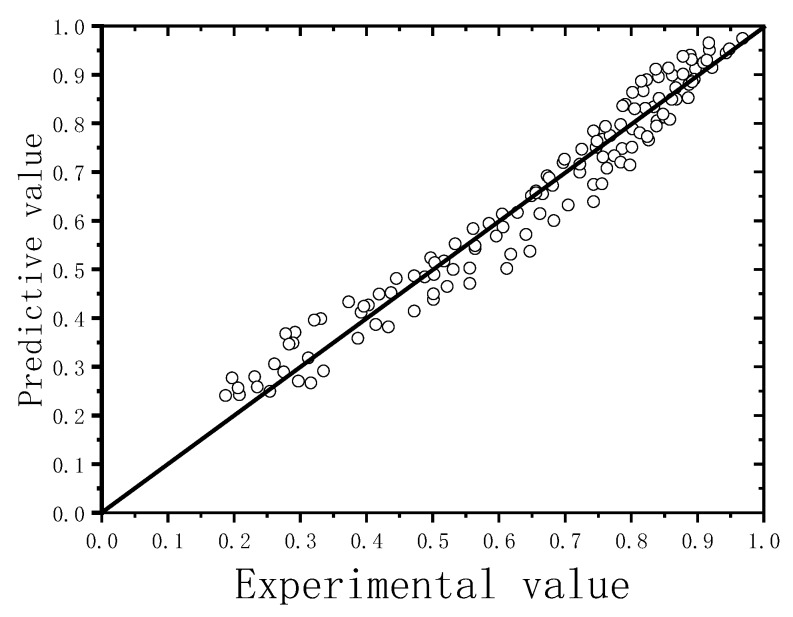
Verification of the kinetic model.

**Table 1 materials-12-03303-t001:** Chemical composition of fly ash (wt. %).

Chemical Composition	Fly Ash
SiO_2_	49.21
Al_2_O_3_	29.46
Fe_2_O_3_	4.36
CaO	2.26
MgO	0.80
TiO_2_	1.21
Na_2_O	0.39
K_2_O	0.28
SO_3_	0.18
Loss on ignition	10.08

**Table 2 materials-12-03303-t002:** CaO absorbed rate of precursors in the hydration reaction at 97 °C.

No.	FA/CA/Na_2_O *	Absorption Rate of CaO (%)									
		2 h	4 h	6 h	8 h	12 h	16 h	20 h	24 h	32 h	40 h
F1	80/20/0	26.1	37.3	49.7	58.5	72.3	76.8	82.1	86.7	89.1	91.7
F2	75/25/0	19.7	32.1	44.5	56.4	64.9	69.9	74.3	80.5	84.1	87.8
F3	70/30/0	20.6	27.8	41.9	50.3	60.5	67.6	72.5	76.1	80.2	83.7
F4	65/35/0	18.7	28.3	39.6	47.2	56.1	65.6	72.2	74.8	78.7	81.5
F2N5	75/25/0.5	23.1	33.1	48.8	53.4	66.6	75.7	80.2	83.4	86.2	88.9
F2N10	75/25/1.0	27.5	39.2	53.1	59.6	68.1	78.6	83.9	86.8	89.7	91.8
F2N15	75/25/1.5	31.2	50.1	56.4	66.2	78.4	83.8	86.1	89.5	94.3	96.2
F3N5	70/30/0.5	23.5	29.2	43.7	51.7	62.8	67.3	74.7	78.4	81.8	85.6
F3N10	70/30/1.0	31.6	43.3	52.2	61.8	70.5	76.3	82.6	85.1	88.7	90.9
F3N15	70/30/1.5	33.5	47.2	55.6	64.1	75.6	80.1	85.8	88.6	92.2	94.8
F4N5	65/35/0.5	20.8	28.9	40.3	50.2	60.6	65.6	69.7	75.6	79.1	82.3
F4N10	65/35/1.0	25.4	38.7	50.1	61.2	68.3	74.3	77.4	81.3	84.2	87.7
F4N15	65/35/1.5	29.7	41.4	55.6	64.7	74.3	79.8	82.4	84.8	89.2	91.4
F3 ^a^	70/30/0	16.6	26.8	33.9	40.3	45.7	49.6	52.5	56.3	60.3	62.7
F3 ^b^	70/30/0	N/A	6.4	N/A	10.1	13.6	16.2	18.7	20.1	N/A	N/A

^a^ The reaction temperature was 57 °C; ^b^ the reaction temperature was 77 °C; * Na_2_O was added as equivalent NaOH and taken as the percentage of the total mass of CaO and fly ash.

**Table 3 materials-12-03303-t003:** Kinetic parameters of the reaction.

No.	Mixed Control					α = 1 − (1 − e^(lnt − lnk)/N^)^3^
	1/N	(ln k)/N	k × 10^2^ (h^−1^)	N	Standard Deviation (%)	
F1	0.61	−2.71	1.29	1.63	2.68	α_1_ = 1 − (1 − e^(lnt − 4.350)/1.63^)^3^
F2	0.65	−2.95	1.07	1.54	3.59	α_2_ = 1 − (1 − e^(lnt − 4.537)/1.54^)^3^
F3	0.63	−2.99	0.88	1.58	3.29	α_3_ = 1 − (1 − e^(lnt − 4.733)/1.58^)^3^
F4	0.64	−3.07	0.85	1.55	3.04	α_4_ = 1 − (1 − e^(lnt − 4.676)/1.55^)^3^
F2N5	0.63	−2.84	1.13	1.58	3.30	α_5_ = 1 − (1 − e(^lnt − 4.483)/1.58^)^3^
F2N10	0.59	−2.63	1.18	1.69	2.43	α_6_ = 1 − (1 − e^(lnt − 4.439)/1.69^)^3^
F2N15	0.57	−2.43	1.43	1.75	2.14	α_7_ = 1 − (1 − e^(lnt − 4.247)/1.75^)^3^
F3N5	0.61	−2.88	0.88	1.64	2.97	α_8_ = 1 − (1 − e^(lnt − 4.733)/1.64^)^3^
F3N10	0.53	−2.46	0.96	1.89	1.79	α_9_ = 1 − (1 − e^(lnt − 4.646)/1.89^)^3^
F3N15	0.54	−2.40	1.19	1.85	1.41	α_10_ = 1 − (1 − e^(lnt − 4.431)/1.85^)^3^
F4N5	0.61	−2.94	0.83	1.63	3.08	α_11_ = 1 − (1 − e^(lnt − 4.791)/1.63^)^3^
F4N10	0.56	−2.62	0.90	1.80	3.39	α_12_ = 1 − (1 − e^(lnt − 4.710)/1.80^)^3^
F4N15	0.54	−2.48	1.02	1.85	3.43	α_13_ = 1 − (1 − e^(lnt − 4.585)/1.85^)^3^
F3 ^a^	0.50	−3.02	0.249	0.50	2.59	α_14_ = 1 − (1 − e^(lnt − 5.995)/1.98^)^3^
F3 ^b^	0.62	−4.59	0.061	0.62	11.88	α_15_ = 1 − (1 − e^(lnt − 7.402)/1.61^)^3^

**Table 4 materials-12-03303-t004:** Value of the kinetic constants and the standard deviation for the model.

No.	k × 10^2^ (h^−1^)		Standard Deviation (%)
	Predictive Value	Experimental Value	
F1	1.28	1.22	3.00
F2	1.06	1.07	0.50
F3	0.92	0.88	2.00
F4	0.82	0.85	1.50
F3 ^a^	0.261	0.249	0.60
F3 ^b^	0.064	0.061	0.15

**Table 5 materials-12-03303-t005:** Value of a with different Na_2_O addition.

No.	k (10^−2^)/h^−1^	a (10^6^)	Average of a (10^6^)
F2N5	1.13	8.78	8.11
F3N5	0.88	7.34	
F4N5	0.83	8.21	
F2N10	1.18	9.39	9.21
F3N10	0.96	8.60	
F4N10	0.90	9.64	
F2N10	1.43	12.57	12.26
F3N10	1.19	12.23	
F4N10	1.02	11.98	

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
