# Peer review of "A Kinetic Study of the Pozzolanic Reaction of Fly Ash, CaO, and Na2O in the Preparation of Fly Ash Belite Cement"

_materials, 2019, doi:10.3390/ma12203303_

Round 1

Reviewer 1 Report

The authors report the kinetic study of the reaction of fly ash/lime granules to prepare belite cement. They report that the use of Na2O improves the hydrothermal reaction and they discuss a kinetic model that could predict the hydrothermal reaction degree.

The experimental results here presented are in good agreement with the data arising from the model, the analysis is exhaustive and, overall, the paper is well presented.

English needs to be revised, because sometimes the discussion is unclear.

Since many other activators have been used in the literature, what are the actual benefits of choosing Na2O? You need to properly introduce Na2O as alkali activator, also by including some references in the introduction. How this study would increase the production efficiency and reduce the production cost?

I include here below are some specific comments:

Line 29: what “wet process” is referred to? Lines 38-40: Actually Na2O has been already reported in other studies in the literature. You need to improve your selection of references and the differences between previous studies should be clarified. For example, ref 12 already reported a study of samples containing different Na2O dosage while varying the calcination temperature. Lines 42-43: if the heat is consumed during the preparation of the materials as granules, is the total amount of heat really reduced or it is just consumed in the previous step? Line 46: fly ash and lime react in stoichiometric ratio? Line 77: AR? Why did not you use Na2O directly? Line 88: why the treatment was made at such low temperature? At line 34 you reported that the synthesis should occur between 97 and 250°C. Please clarify why you refer to CaO as “absorbed” The details about the experiments are missing (The instruments you used, the way you calculated the mass losses from the TG curves..). According to the journal guidelines, the you should include the main methods used. Why did not you show the experimental results? Maybe you can include some curves, also in the SI… Lime 92: 500 mg? Are you sure? Lines 103-106: Isn’t is obvious? At high FA/CA ratio there is less CaO Lines 106-109: You should justify this speculations Table 2: I am unclear why it is important to look at both N and k factors. Why N and k do not show the same trend? Line 118: Did you measure the particle size distribution? You say that you have 8-12 mm particles, and you assume that they are at a constant size? Table 3: you should include the error bars Figure 1: how did you study the simulated curves? Figure 2: what about the correlation coefficient of this fitting curve? Are the dots the actual experimental point? Line 155: how can you say that the reaction path and steps do no change? Line 161: you discuss first figure 4 and later figure 3 Lines 179-182: you should further discuss the speculations you made, that should be corroborated by evidence. Figure 6: why don’t you show a table analogous to table 4, with the new predicted values calculated using equation 9 instead of equation 6? It would allow to numerically see if the standard deviation is now reduced, including the correlation factor. Line 212-216: what do you mean with crystallization rate of C-S-H?? I am sure you now that C-S-H binder gel phase is a well know amorphous phase. Figure 5, not 6 Ref 14 is not correctly cited check again typos and english

Reviewer 2 Report

Preparation of specimens need more information. How the mix designs, temp and time are selected? Line 104-109: If the observations and remarks are observed in other studies, cite them to support your findings. Section 4. Line 212, hydratation accelerated in the presence of Na2O. It is very general remark. Authors can provide more quantitative results. Similarly, In the conclusion some scientific points are missing. dosage of Na2O > 1% need a scientific reason for complete understanding. Experiments accuracy and repeatability can be defined in the method section. Some recent application of fly ash-based material can be added to strengthen the sustainability aspect described in the introduction.  (Improving the 3D printability of high volume fly ash mixtures via the use of nano attapulgite clay; Synthesis and calorimetric study of hydration behavior of sulfate‐rich belite sulfoaluminate cements with different phase composition)
